# Application of ^12^C^6^ Heavy Ion-Irradiated BHK-21 Cells in Production of Foot-and-Mouth Disease Vaccine

**DOI:** 10.3390/vetsci12020167

**Published:** 2025-02-13

**Authors:** Xiangdong Song, Shiyu Tao, Fanglan An, Xiaoming Li, Jingcai Yang, Yan Cui, Xuerong Liu

**Affiliations:** 1College of Veterinary Medicine, Gansu Agricultural University, Lanzhou 730070, China; 107332214007@st.gsau.edu.cn; 2China Agricultural Vet Biology and Technology Co., Ltd., Lanzhou 730046, China; taosy1989@126.com (S.T.); 18198093582@163.com (F.A.); 17389315860@163.com (X.L.); doniesong@gmail.com (J.Y.); 3Lanzhou Veterinary Research Institute, Chinese Academy of Agricultural Sciences, Lanzhou 730046, China

**Keywords:** ^12^C^6^ heavy ion, FMDV, vaccine, cell suspension culture

## Abstract

The foot-and-mouth disease (FMD) vaccine is a critical tool for preventing FMD outbreaks. However, current production methods face challenges in achieving sufficient capacity. This study investigates the use of heavy ion irradiation on wild-type BHK-21 cells to develop a mutant cell line, BHK-7, aimed at enhancing vaccine production efficiency. The goal is to establish an innovative method for creating an FMD vaccine cell line. The research findings indicate that the mutant BHK-7 cell line exhibits a significant increase in the content of FMD antigen 146S and demonstrates stable replication after suspension adaptation. Additionally, the analysis of the culture substrate requirements reveals that glutamine plays a crucial role in FMD virus (FMDV) replication within this cell line. The conclusion is that the heavy ion irradiated cell line has the potential to improve production efficiency and reduce costs.

## 1. Introduction

The outbreak of FMD poses a significant threat to the robust development of animal husbandry and public health safety. Over an extended period, scholars worldwide have dedicated substantial efforts to conducting extensive and in-depth research on this issue [1]. In recent years, antiviral research has emerged as a promising new frontier, achieving remarkable progress. Traditional antiviral strategies have primarily centered on drug and vaccine development. Regarding drug development, scientists have been actively seeking compounds capable of directly inhibiting FMDV activity [2]. In vaccine development, more efficient and safer production technologies are continuously being explored [3]. However, with the deepening understanding of viral replication mechanisms, it has become increasingly evident that relying solely on traditional methods is insufficient. The importance of exploring new targets and innovative strategies has thus gained prominence. Heavy ion irradiation technology, as a promising novel approach, has garnered considerable attention in the biological field. Numerous studies indicate that radiation can induce cellular mutations that may affect viral replication [4]. With the deepening understanding of viral replication mechanisms, traditional research methods have gradually revealed their limitations. In recent years, heavy ion irradiation technology, as an emerging biophysical approach, has garnered significant attention in the field of biology. Studies have shown that heavy ion irradiation can induce specific genetic mutations and epigenetic modifications in cells, thereby influencing the regulation of cellular signaling pathways and altering the viral replication environment [5,6,7]. However, research on the biological effects of heavy ion irradiation on FMDV and its host remains limited, particularly regarding the molecular mechanisms underlying the interaction between heavy ions and the virus–host system. This presents an important direction for future research.

At the molecular level, heavy ion irradiation has been demonstrated to modulate the function of *MAVS* (mitochondrial antiviral signaling protein) and influence viral replication by activating key signaling pathways, such as mTOR (mechanistic target of rapamycin), *NF-κB* (nuclear factor kappa B), and *COX-2* (cyclooxygenase-2) [8,9,10]. These findings are closely related to the replication mechanisms of FMDV. For instance, FMDV infection promotes the translocation of mTOR to lysosomes, enhances the interaction between *mTOR* and *Rheb,* and activates the *PI3K/AKT/TSC2/Rheb/mTOR/p70S6K1* signaling pathway to facilitate viral replication [8]. Additionally, the FMDV 2C protein interacts with *Nmi* (*N-myc* and *STAT* interactor) to negatively regulate virus-induced type I interferon production while inducing apoptosis to promote viral replication [11,12]. Notably, *MAVS*, as a crucial adaptor protein in the *RLR* (RIG-I-like receptor) pathway, plays a pivotal role in antiviral immunity. Research has revealed that the FMDV structural protein VP3 interacts with *MAVS*, inhibiting its expression by interfering with mRNA synthesis, thereby reducing the host’s antiviral capacity and creating favorable conditions for viral replication [13].

Based on these findings, heavy ion irradiation may modulate the normal functions of host cells, inducing intracellular environmental changes conducive to FMDV replication, thereby promoting viral replication within cells. This characteristic offers new insights for screening cell lines with high replication efficiency and opens new avenues for viral culture techniques in vaccine production.

Regarding vaccine production, cell suspension culture technology is gradually showing significant advantages. By employing suspension culture and domesticating BHK-21 cells after heavy ion irradiation, we can achieve efficient and stable vaccine production. Suspension culture provides a larger cultivation space, which facilitates rapid cellular proliferation and substantial viral replication. Compared to traditional adherent culture methods, suspension culture not only enhances production efficiency but also reduces costs. In this study, BHK-21 cells were irradiated. After screening and evaluation, the mutant somatic cell line BHK-7 was finally obtained. Under suspension culture conditions, the viral productivity of the BHK-7 cells was markedly higher than that of wild-type BHK-21 cells, providing robust support for large-scale foot-and-mouth disease vaccine production. Additionally, this research offers valuable methodologies and strategies for the development of other vaccine cell lines.

## 2. Materials and Methods

### 2.1. Cells and Viruses

The wild-type BHK-21 cell line (ATCC CCL-10) was utilized for cell culture. The FMDV O/MYA98 strain and the inactivated FMDV antigen were obtained from China Agricultural Vet Biology and Technology Co., Ltd., Lanzhou, China

### 2.2. Optimization of Irradiation Parameters for ^12^C^6^ Heavy Ions

BHK-21 cells were cultured in a 35 mm cell culture dish at a density of 1 × 10^5^ cells/mL. They were induced by a carbon ion beam accelerated by the Lanzhou Heavy Ion Research Institute (HIRFL) of the Institute of Modern Physics, Chinese Academy of Sciences. The ion beam is (^12^C^6^), and the initial energy of the ion beam is 80.55 MeV/u, with doses set at 0 Gy, 5 Gy, 10 Gy, 15 Gy, 20 Gy, and 25 Gy, respectively (Table 1). Six replicates were established for each dose group. After irradiation, the medium was refreshed. Then, an inactivated *O/MY/A98* complete FMDV antigen (2 ng/mL) was added to stimulate the cells. The cells were then incubated at 37 °C in a 5% CO_2_ atmosphere for 24 h. After that, cell viability was assessed and mortality was determined. Dead or apoptotic cells were removed, and the culture was continued for another 48 h. Subsequently, the cells were diluted to a uniform concentration and seeded into 6-well plates. FMDV infection was carried out at a multiplicity of infection (MOI) of 0.1 for 16 h, with three replicates for each group. The cells and culture medium were subjected to three freeze–thaw cycles and stored overnight at −20 °C. The content of intact FMDV particles (146S) was measured to determine the appropriate dose of heavy ion irradiation for induction.

### 2.3. Screening and Characterization of Monoclonal Cell Lines After ^12^C^6^ Heavy Ion Irradiation

Following irradiation with ^12^C^6^ heavy ions, the cells were incubated at 37 °C and 5% CO_2_ for 48 h. During this period, cell viability was rigorously monitored, and non-viable cells were promptly removed. Surviving cells were subsequently screened using a limiting dilution method. Specifically, the cell suspension was serially diluted and transferred to a 96-well plate. A 20× microscope was used to confirm that each well contained a single clone. If multiple cells were observed in any well, the dilution ratio was adjusted accordingly until all wells met the criteria for monoclonality. The 96-well plates containing monoclonal cells were then cultured in a humidified incubator set at 37 °C with 5% CO_2_. Cell growth was closely monitored, and when confluence reached approximately 90%, the monoclonal cells were carefully transferred to 6-well plates for further expansion. Culturing was continued for at least 10 passages to stabilize the characteristics of the monoclonal cells.

### 2.4. Evaluation of FMDV Replication Efficiency in Mutant BHK-21 Cells Induced by ^12^C^6^ Heavy Ions

Following the screening of monoclonal cells, both the mutant cell line and the control BHK-21 cells were adjusted to an equal concentration, seeded into a 96-well cell culture plate, and incubated in a humidified 5% CO_2_ atmosphere until they reached approximately 90% confluence. Subsequently, both cell types were infected with FMDV at an MOI of 0.1. After a 16 h incubation period, the cells and culture medium underwent three freeze–thaw cycles to assess the levels of intact virions (146S).

### 2.5. TCID_50_

BHK-21 cells were seeded in 96-well plates until they reached 90% confluence. Virus samples derived from BHK-7 and control BHK-21 cells were serially diluted from 10^−1^ to 10^−8^ and placed in a separate plate. Each well was inoculated with 100 μL of the diluted samples, and the plates were incubated at 37 °C for 1 h. After incubation, the inoculum was removed, and the cells were cultured in DMEM (Dulbecco’s modified eagle medium, Gibco C11995500BT) containing 1% FBS (FBS; Gibco 10099141) for 72 h. The TCID_50_ values were determined using the Reed–Muench method.

### 2.6. Viral Replication Was Quantified by qRT-PCR

BHK-21 cells served as the control group, whereas mutant BHK-7 cells were seeded in a 6-well plate to ensure equal cell numbers across all wells (to standardize cell concentration among experimental groups). Upon reaching approximately 90% confluence, the cells were washed three times with PBS. They were then incubated with FMDV at a multiplicity of infection (MOI) of 1.0 for 1 h at 37 °C, followed by washing with PBS and culturing in 2 mL of DMEM supplemented with 1% FBS. Uninfected cells served as an additional control group. The cells were subsequently cultured in a humidified 5% CO_2_ incubator at 37 °C for 16 h. After this period, the culture medium and cells were harvested. Samples underwent three freeze–thaw cycles and were stored overnight at −20 °C. Gel filtration chromatography and column molecular sieve techniques [14] were employed to quantify the total concentration of FMDV virus particles (146S) in the samples. Total RNA was extracted from FMDV-infected cell culture supernatants using TRIzol reagent (Invitrogen, Carlsbad, CA, USA, 15596026) and reverse-transcribed into cDNA using the PrimeScript^™II^ First Strand Synthesis Kit (RR047A, Takara, Beijing, China) according to the manufacturer’s protocol. The expression levels of the FMDV gene (FMDV-3D-F: GAACACATTCTTTACACCAGGAT, FMDV-3D-R: CATATCTTTGCCAATCAACATCAG) and the internal reference gene GAPDH (GAPDH-F: ATGGCCTTCCGTGTTCCTAC, GAPDH-R: GCCTGCACCACCTTCTT) were subsequently analyzed. All the experiments were repeated at least three times, and relative mRNA expression levels were calculated via the threshold cycle (2^−△△Ct^) method.

### 2.7. Domestication of BHK-7 Cells in Suspension Culture

Select mutant BHK-7 cells that have a growth fusion degree of 90% and a viability exceeding 95%. After trypsin digestion, prepare a cell suspension with a concentration of 0.5 × 10^6^ cells/mL. Initially, perform static culture for 48 h using a medium mixture of 20% suspension medium and 80% DMEM supplemented with 5% FBS (the suspension medium is provided by China Agricultural Vet Biology and Technology Co., Ltd., Lanzhou, China). Subsequently, we gradually increased the proportion of suspension medium to 30% and then 50%, with each increase followed by a 48 h static culture period to allow gradual adaptation to the suspension medium. After culturing for 48 h with a suspension medium to DMEM ratio of 50:50, we digested the cells with trypsin, selected cells with viability greater than 95%, transferred them to a fully suspension medium at an initial density of 0.5 × 10^6^ cells/mL, and cultured them under conditions of 5% CO_2_ and 100 rpm. Continuous culture was maintained until the cells stabilize. When the cell density reached 3.0 × 10^6^ cells/mL and the cell aggregation rate was less than 10%, this indicated that BHK-7 cells have successfully adapted to the suspension medium and they were designated as BHK-7-FC.

### 2.8. BHK-7-FCcell Suspension Proliferation Experiment

BHK-7-FC cells were inoculated at an initial density of 0.5 × 10^6^ cells/mL and cultured continuously for 10 passages. Samples were collected every 24 h to monitor cell viability and density, thereby assessing the stability of cell proliferation.

### 2.9. Analysis of Metabolites in BHK-7-Fc Cell Suspension Medium

BHK-7-FC cells were cultured at an initial density of 0.5 × 10^6^ cells/mL and infected with FMDV at a multiplicity of infection (MOI) of 0.001. Samples were taken every 3 h to determine the consumption levels of key substrates, including glutamine (GLN), lactic acid (LAC), glucose (GLUC), and ammonium ion (NH_4_^+^).

### 2.10. Evaluation of the Proliferation Stability of BHK-7-FC Cells for FMDV

BHK-7-FC cells were cultured continuously for 10 passages in 250 mL cell culture flasks following the standard BHK-7-FC cell culture protocol. Every three passages, cells with a density greater than 5.0 × 10^6^ cells/mL and viability exceeding 95% were selected. After settling at 4 °C for 1 h, cells were centrifuged at 1000 rpm for 10 min. Subsequently, 80 mL of FMDV maintenance solution (MOI = 0.01) was added to the flask. After 16 h of incubation, the culture supernatant was collected and analyzed for 146S antigen content and TCID_50_.

### 2.11. Statistical Analysis

The experimental data were analyzed through Student’s two-tailed unpaired *t*-tests and two-way analysis of variance (ANOVA) with the aid of GraphPad Prism 9. *p*-values of <0.05 were statistically significant.

## 3. Results

### 3.1. Optimization of ^12^C^6^ Heavy Ion Irradiation Parameters

Following the induction treatment of BHK-21 cells with^12^C^6^ heavy ions at doses of 0 Gy, 5 Gy, 10 Gy, 15 Gy, 20 Gy, and 25 Gy, it was observed that as the ^12^C^6^ heavy ion dose gradually increased, the viability of the induced cells exhibited a significant downward trend. Specifically, when the^12^C^6^ heavy ion dose reached 25 Gy, cell viability decreased to only 10%. In Figure 1a, the changes in BHK-21 cell viability induced by different doses of heavy ions are clearly demonstrated. Subsequently, the heavy ion-irradiated cells were seeded into six-well cell culture plates. Once the cells reached 90% confluence, they were infected with FMDV at a multiplicity of infection (MOI) of 0.1. After a continuous infection period of 16 h, the content of intact 146S viral particles was measured. The results indicate that as the heavy ion dose increased, the viral replication level initially improved, reaching its peak at a dose of 15 Gy. Beyond this point, the viral replication level remained relatively stable. As shown in Figure 1b, these data clearly demonstrate that at lower doses, specifically 5 Gy, there was a significant inhibition of viral replication. Conversely, doses greater than 15 Gy resulted in a slight promotion of viral replication. Based on these experimental findings, three dose groups, 5 Gy, 10 Gy, and 15 Gy, were selected for subsequent experiments.

### 3.2. Screening and Characterization of Mutant Cells Following ^12^C^6^ Induction in BHK-21 Cells

The 146S antigen content serves as a critical index for measuring antigen levels in FMD vaccine production. BHK-21 cells were subjected to irradiation at three dose levels: 5 Gy, 10 Gy, and 15 Gy. After 48 h of irradiation, surviving cells were selected using the limiting dilution method, with 20 monoclonal cell lines randomly chosen from each dose group. Once these cells reached stable growth, they, along with control BHK-21 cells, were diluted to a concentration of 1 × 10^6^ cells/mL. Subsequently, FMDV infection was performed at a multiplicity of infection (MOI) of 0.1 for 16 h, followed by quantification of the 146S antigen content. As illustrated in Figure 1c, mutant cell lines, such as BHK-7, 15B9, 15B11, c8, 5G3, 5G7, and 20E9K1, exhibited significantly higher 146S antigen levels compared to the control BHK-21 cells, with BHK-7 showing the most pronounced increase. To further investigate the replication efficiency of FMDV in BHK-7 cells, an infection experiment was conducted at an MOI of 1.0 for 16 h. The results, shown in Figure 2a,b, demonstrate that the mutant cells had significantly enhanced FMDV replication, both in terms of complete virus particle (146S) content and viral gene expression levels. Additionally, the TCID_50_ of the FMDV-infected BHK-7 cells was evaluated at 4h, 8h, and 16h post-infection. As depicted in Figure 2c, BHK-7 consistently exhibited higher TCID_50_ values than BHK-21 at all time points. Notably, the growth rates of BHK-7 and BHK-21 cells were comparable, as shown in Figure 2d. These findings suggest that BHK-7 is a cell line that effectively promotes FMDV replication.

### 3.3. Adaptation of BHK-7 Cells to Suspension Culture

Given that cell adaptation to suspension culture is a gradual process, we progressively increased the proportion of the suspension medium by mixing it with the adhesive medium DMEM in varying ratios. Following acclimatization, as illustrated in Figure 3a, mutant BHK-7 successfully adapted to suspension culture and exhibited a growth state comparable to that of the control BHK-21 suspension cells (a cell line used to produce the foot-and-mouth disease vaccine by China Agricultural Veterinary Biology Ltd. Lanzhou, China). Figure 3b demonstrates that the BHK-7-FC cells maintained stable growth and viability over 10 passages. Figure 3c,d indicates that the cell density reached 6 × 10^6^ cells/mL on day 3, with the cell viability exceeding 95%. These results confirm that BHK-7 has successfully undergone suspension adaptation and has been designated as BHK-7-FC for subsequent studies.

### 3.4. Analysis of Metabolites in BHK-7-FC Suspension Culture

Key substrates in cell suspension media, including glutamine (GLN), lactate (LAC), glucose (GLUC), and ammonium ions (NH_4_^+^), play a crucial role in viral infection and cell growth, serving as critical indicators of the quality of suspension media. The consumption and accumulation of these substrates are directly linked to the energy supply required for cell growth and viral replication. Upon examination, the changes in NH_4_^+^, LAC, and GLUC levels in BHK-7-FC cells were not significantly different when compared to control BHK -21 cells (Figure 4b–d). However, GLN levels were markedly reduced 12 h post-infection with FMDV in BHK-7-FC cells (Figure 4a). The possible reason is that heavy ion irradiation induces the GLN-related metabolic functions of mutant cells, which results in the upregulation of GLN metabolism, which is consistent with the findings of many previous studies [15]. Based on our research results, where BHK-7 shows a significantly enhanced consumption of GLN, we speculate that GLN might be one of the important substrates for the replication of FMDV in mutant BHK-7. Therefore, we conducted the subsequent experiment of adding GLN to the culture medium, suggesting that GLN may play a pivotal role in FMDV replication. Subsequently, various concentrations of GLN were added to the BHK-7-FC cell suspension medium, and the levels of 146S particles were measured post-FMDV infection to evaluate the effect of GLN on viral replication. As shown in Figure 4e,f, the addition of 1 mmol/L GLN to the medium resulted in a significant increase in 146S particle content. When 1 mmol/L GLN was added simultaneously to both BHK-21 and BHK-7-FC cell culture media, the results indicated a substantial increase in 146S particle content in BHK-7-FC cells following GLN supplementation, while no significant difference was observed in BHK-21 cells infected with FMDV. This suggests that the surface mutant BHK-7-FC may enhance FMDV replication by altering metabolic pathways and increasing energy consumption. The transmembrane glycoproteins in the viral envelope mediate membrane fusion and viral entry by binding to host cell receptors. Infected host cells substantially upregulate glutamine metabolism to generate ATP and precursors for macromolecular synthesis (such as nucleotides and lipids), thus supporting the assembly and release of progeny viruses [16]. Moreover, glutamine serves as a precursor for the synthesis of nucleotides, amino acids, and lipids, all of which are essential to produce new viral components. Studies have demonstrated that glutamine is implicated in the replication of multiple viruses and is involved in cellular metabolism. For instance, Newcastle disease virus (NDV) facilitates the catabolism of glutamine into the tricarboxylic acid cycle. Solute carrier family 1 member 3 (SLC1A3) is a crucial regulator of glutamine metabolism. Silencing SLC1A3 reduces the levels of glutamate and glutamine catabolism and promotes the replication of NDV. Likewise, findings from studies in BHK cells indicate that the replication of Sendai virus (HVJ) necessitates glutamine. The absence of glutamine in the culture medium inhibits the release of HVJ and the synthesis of envelope proteins [17].

In summary, supplementing the suspension medium with 1 mmol/L GLN can significantly improve FMDV replication efficiency. Additionally, the inclusion of GLN in BHK-7-FC culture media can further enhance viral production.

### 3.5. Evaluation of the Stability of the BHK-7-FC

To comprehensively and accurately assess the replication performance of BHK-7-FC cells against FMDV after suspension domestication, we conducted a serial passage experiment over 10 consecutive generations. During this process, the levels of 146S particles and TCID_50_ were measured every three passages. The results, following a meticulous analysis, clearly demonstrated that the mutant BHK-7-FC cells exhibited high stability in replicating FMDV post-suspension domestication. Specifically, both the 146S particle count and TCID_50_ levels remained stable throughout continuous passages, strongly indicating the robustness of BHK-7-FC cell replication against FMDV post-suspension acclimation (Figure 5a,b). Further analysis revealed that the 146S antigen production by the BHK-7-FC cells was consistently and significantly higher than that of the control BHK-21 cells. Similarly, there were notable differences in their TCID_50_ levels. These findings were consistent with those observed in the mutant BHK-7 cells studied previously.

The results confirm that the suspension adaptation process successfully preserves both the viral sensitivity and genetic stability of this engineered cell line. Notably, the 146S yield obtained in this study exceeds the highest reported values for suspension-adapted BHK-21 systems. When compared to the results of previous studies, it shows a significant increase in the 146S yield [18,19]. This indicates that BHK-7-FC cells have the potential to substantially cut down vaccine production costs. This can be achieved by reducing the cell culture volume and the requirements for downstream purification. These findings firmly establish BHK-7-FC as a robust platform for industrial-scale FMDV production. They provide both theoretical and practical bases for future vaccine development, thereby paving the way for more efficient and cost-effective vaccine manufacturing processes.

## 4. Discussion

The outbreak and spread of foot-and-mouth disease (FMD) have inflicted substantial economic losses on the global livestock industry and posed a significant threat to animal health and safety. Consequently, the implementation of effective prevention and control measures is crucial for safeguarding the development of animal husbandry. Currently, inactivated FMD vaccines play a pivotal role in disease prevention, with their immunogenic efficacy directly impacting both animal health and the stability of the livestock sector. The concentration of 146S antigen in the vaccine is critical for eliciting an immune response [20,21], indicating that enhancing the 146S antigen content represents a key strategy for improving the quality of FMD vaccines.

In this study, wild-type BHK-21 cells were irradiated with 12C6 heavy ions, and the irradiation parameters were meticulously optimized. The results showed that cell viability decreased significantly as the heavy ion dose increased. This reduction in viability can be attributed to the extensive damage inflicted by high doses of heavy ions on cellular structures, thereby disrupting normal metabolic processes and compromising cell survival. However, within a specific dose range, viral replication levels exhibited a gradual increase. Specifically, FMDV inhibition was observed at doses below 15 Gy, while doses exceeding 15 Gy slightly promoted FMDV replication. These findings align with the research by K. Lumniczky et al., which indicates that high doses primarily exert immunosuppressive effects, whereas low doses modulate immune responses [22]. Lower doses of heavy ions may inhibit FMDV replication by activating cellular immune functions, while higher doses promote viral replication through immunosuppression, likely due to the complex regulation of cellular immunity. At lower doses, cells may initiate various immune defense mechanisms, enhancing their resistance to viral infection. Conversely, at higher doses, severe immunosuppression facilitates viral replication within the cells.

In other areas of virology research, relevant studies have previously been published. One study demonstrated that ultraviolet (UV) irradiation of BHK-21 cells can enhance their susceptibility to rabies virus [23]. The research by P. E. Ceccaldi et al. further confirmed that ionizing radiation can increase the infection intensity and prolong the duration of rabies virus infection in mouse brains through an immunosuppressive mechanism in a dose-dependent and reversible manner. Specifically, higher radiation doses resulted in more severe viral infections and longer infection durations in the mouse brains [24]. This phenomenon exhibits a similar trend to the results observed in this study. In our study, as the induced dose of heavy ions increased, the replication level of FMDV gradually rose (Figure 1b). Additionally, it is noteworthy that irradiation techniques, including heavy ions and rays, have been widely applied in the inactivation of viruses for vaccine development [25,26]. Heavy ions possess unique physical and biological characteristics. Compared with traditional rays, they can deposit more concentrated and intense energy in organisms, thereby triggering specific biological changes in cells. It can induce a high mutation rate, a wide mutation spectrum, and excellent mutant stability [27]. The generated mutants can increase the possibility of discovering new virus–host interaction mechanisms. Moreover, this approach avoids the limitations that may be caused by the purposeful manipulation of specific genes. It is closer to the variation in the natural state and can produce multiple different mutation types, providing richer materials for research [28,29]. These advantages make heavy ion irradiation a highly promising research tool.

The content of FMDV 146S in the mutant BHK-7 cells obtained by heavy ion irradiation-induced screening was significantly increased. These findings unequivocally demonstrate that heavy ion irradiation can effectively induce mutant cells with enhanced viral replication capacity. The development of these mutant cells offers a novel cellular resource to produce FMD vaccines. Following adaptation to suspension culture, we observed that the growth rate of mutant BHK-7 cells was comparable to that of control BHK-21 cells. This suggests a feasible method for large-scale vaccine production. Suspension culture confers several advantages, including the provision of a larger culture volume, facilitation of rapid cell proliferation, and substantial viral replication. Additionally, this culture method can enhance production efficiency and reduce manufacturing costs [30]. In the metabolic analysis of BHK-7-FC suspension cultures, it was observed that glutamine (GLN) may play a pivotal role in FMDV replication. The addition of an appropriate concentration of GLN can enhance the replication efficiency of FMDV, providing a foundation for optimizing medium composition. By examining key substrates, such as GLN, lactate (LAC), glucose (GLUC), and ammonium (NH_4_^+^), in the cell culture medium [31,32], we observed that changes in NH_4_^+^, LAC, and GLUC levels in BHK-7-FC were not significantly different compared to the control BHK-21 cells. However, GLN levels decreased markedly 12 h post-infection with FMDV, indicating a strong correlation between GLN consumption and viral replication. GLN likely plays a critical role in energy provision or biosynthesis during viral replication. Supplementing the medium with an optimal amount of GLN can enhance FMDV replication efficiency, thereby providing a crucial basis for medium optimization. Adjusting GLN concentrations in the medium can improve viral replication efficiency, thus increasing vaccine antigen production. The results of ten successive generations of passage experiments demonstrated that the mutant BHK-7 exhibited stable replication of the FMDV virus following adaptation to suspension culture. This indicates that the mutant cells induced by heavy ion irradiation possess excellent genetic stability and are suitable for long-term vaccine production. To further evaluate the stability of the mutant cells, we conducted ten successive generations of passages and monitored changes in the levels of 146S particles and TCID_50_ every three generations. The findings revealed that the replication of FMDV in mutant BHK-7 remained stable after suspension adaptation. This suggests that the mutant cells induced by heavy ion irradiation exhibit robust genetic stability and can consistently replicate the virus over extended passages. Such stability is crucial for vaccine production, ensuring consistent quality and efficacy.

In summary, this study successfully developed a mutant somatic cell line BHK-7 that enhances FMDV replication through the application of heavy ion irradiation technology. The research further explored its potential in vaccine production via suspension culture domestication and metabolic analysis. This work offers novel insights and methodologies for FMD vaccine development, potentially increasing production efficiency and reducing costs, thereby significantly contributing to the sustainable development of global animal husbandry. Additionally, this study provides valuable references for the development of other vaccine cell lines and broadens the application scope of heavy ion irradiation technology in vaccine research. Future research could delve into the mechanisms of heavy ion irradiation-induced cell mutation, optimizing irradiation parameters and screening methods to enhance mutant cell performance and stability. Moreover, integrating other biotechnological approaches, such as gene editing, may further refine the cell line, improving vaccine quality and efficacy. Concurrently, stringent quality control and safety oversight of the vaccine production process are essential to ensure vaccine safety and effectiveness, providing robust support for the healthy development of the global livestock industry.

## Figures and Tables

**Figure 1 vetsci-12-00167-f001:**
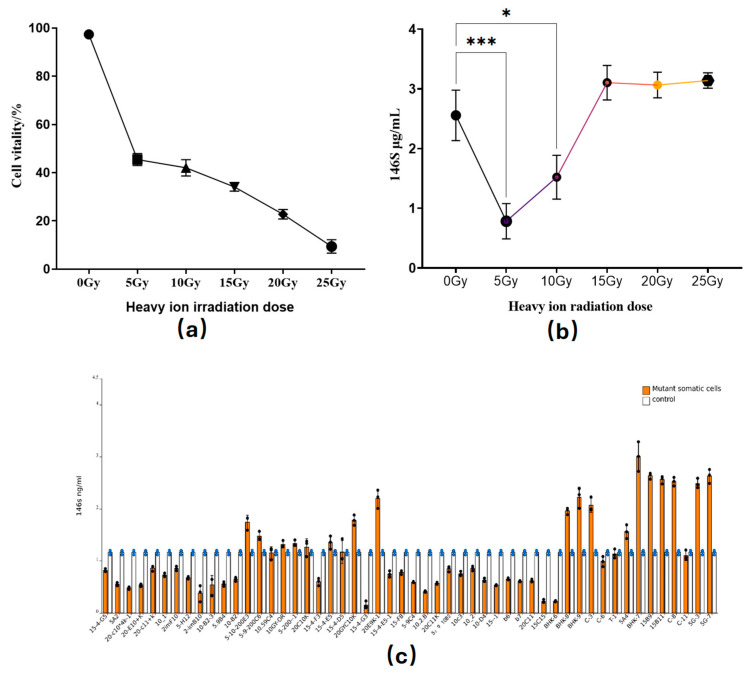
(**a**) BHK-21 cells were induced with 12C6 heavy ions at doses of 0 Gy, 5 Gy, 10 Gy, 15 Gy, 20 Gy, and 25 Gy. Cell viability was assessed via trypan blue staining and detected via Count STAR (Countstar Rigel S2, RY074B2001), and mortality rate was calculated. Data represent means and standard errors from six independent experiments. (**b**) After BHK-21 cells were induced with different doses, foot- and -mouth disease virus (FMDV) was used to infect cells at infection multiplicity (MOI) of 0.1 for 16 h. (**c**) After researchers exposed BHK-21 cells to irradiation doses of 5 Gy, 10 Gy, and 15 Gy, mutant cells obtained through monoclonal screening were infected with FMDV at multiplicity of infection (MOI) of 0.1 for 16 h, resulting in alterations in antigen content, specifically in the 146S component. Experiments were conducted in triplicate as biological replicates. Data are presented as means ± SDs from three independent experiments and were analyzed via Student’s two-tailed unpaired *t*-tests. *, *p* < 0.05; ***, *p* < 0.001. The orange bar graph represents the 146s content after FMDV infection in the mutant cell group, and the white bar graph represents the 146s content after FMDV infection in control BHK-21 cells.

**Figure 2 vetsci-12-00167-f002:**
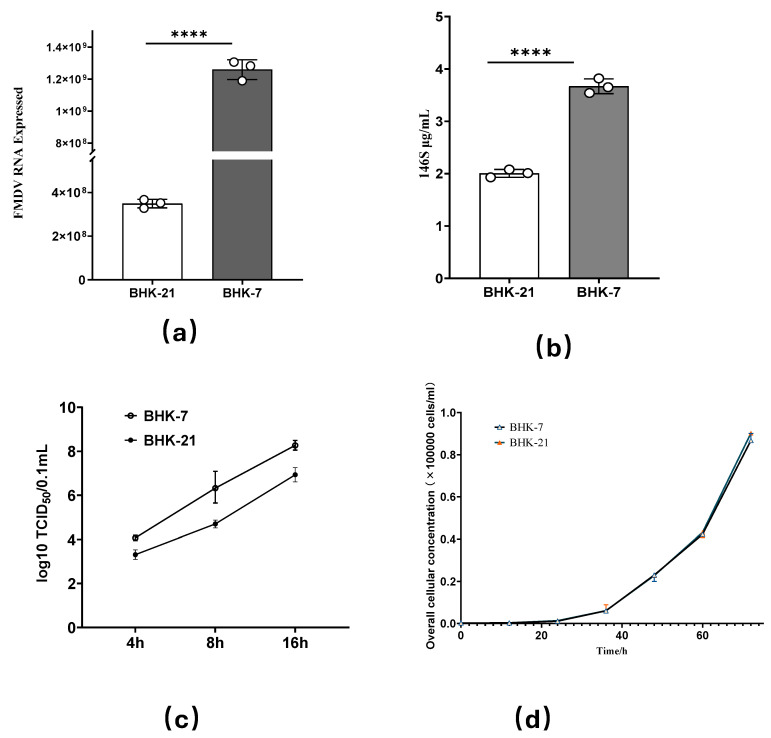
(**a**) Shows expression level of FMDV genes in mutant BHK-7 and control BHK-21 cells infected with FMDV at multiplicity of infection (MOI) of 0.1 for 16 h. (**b**) Shows change in 146S antigen content in mutant BHK-7 and control BHK-21 cells infected with FMDV at multiplicity of infection (MOI) of 0.1 for 16 h. (**c**) Shows change in TCID50 after mutant BHK-7 and control BHK-21 cells are infected with FMDV. (**d**) Shows growth curves of mutant BHK-7 and control BHK-21 cells. Experimental data are all from three biological replicates. Data are presented as means ± SDs from three independent experiments and were analyzed via Student’s two-tailed unpaired *t*-tests.; ****, *p* < 0.0001.

**Figure 3 vetsci-12-00167-f003:**
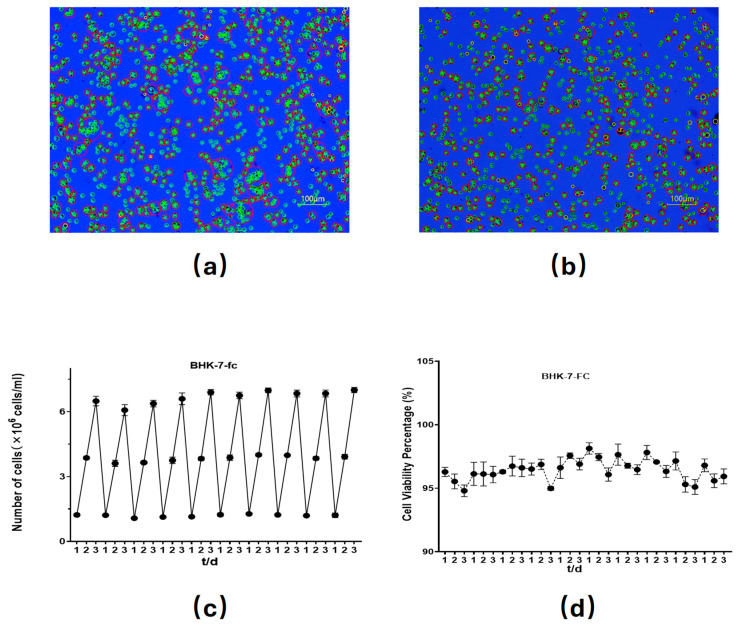
(**a**) BHK-7-FC suspension cells were stained with trypan blue for viability assessment and bright-field imaging. (**b**) BHK-21 suspension cells were stained with trypan blue for viability assessment and bright-field imaging. (**c**) Proliferation curve of BHK-7-FC cells in suspension culture is presented. (**d**) Short-term viability of BHK-7-FC cells in suspension culture over time was evaluated. Data are presented as means ± SDs from three independent experiments. Red circles represent clumped cells.

**Figure 4 vetsci-12-00167-f004:**
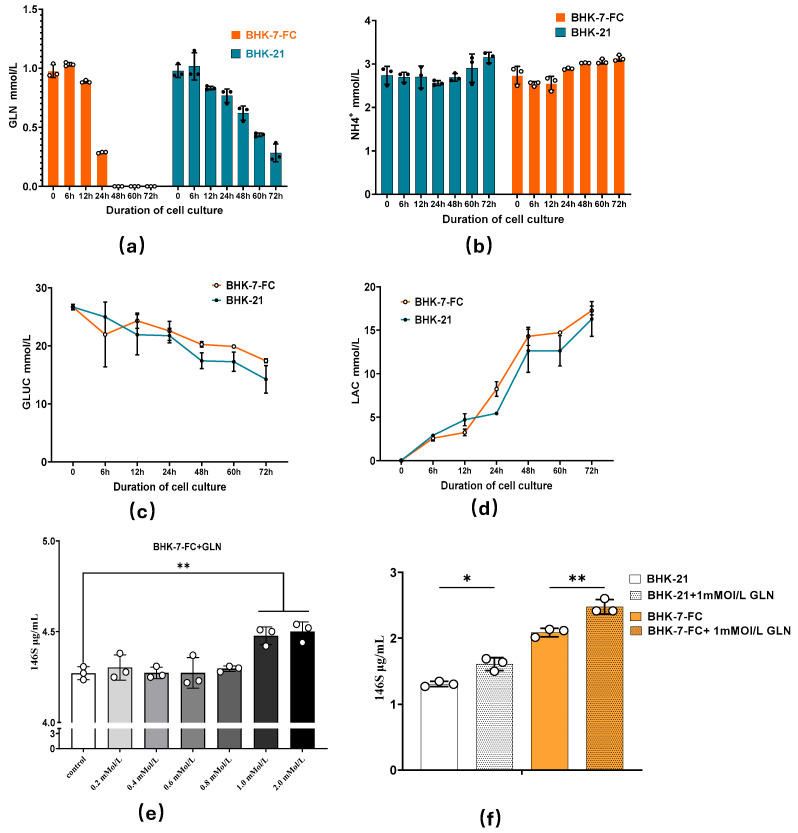
(**a**) Temporal variations in GLN content following FMDV infection in BHK-7-FC suspension cultures. (**b**) Changes in NH_4_^+^ levels at various time points post-FMDV infection in BHK-7-FC suspension cultures. (**c**) Temporal changes in LAC content following FMDV infection in BHK-7-FC suspension cultures. (**d**) Variations in GLUC content over time after FMDV infection in BHK-7-FC suspension cultures. (**e**) Effect of supplementing different concentrations of GLN on 146S antigen content 16 h post-FMDV infection; (**f**) Impact of adding 1 mmol/L GLN to both BHK-7-FC and control BHK-21 cells on 146S antigen content 16 h post-FMDV infection. All experiments were conducted with three biological replicates. Data are presented as means ± SDs from three independent experiments and were analyzed via Student’s two-tailed unpaired *t*-tests. *, *p* < 0.05; **, Black and white dots represent the scatter points on the column, indicating the specific data of each sample within the group.

**Figure 5 vetsci-12-00167-f005:**
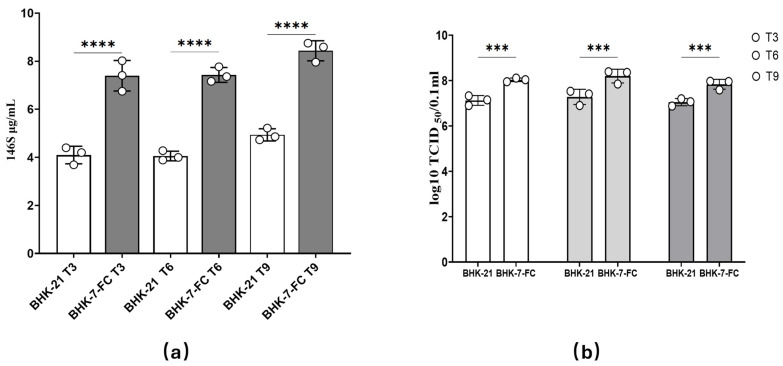
(**a**) The BHK-7-FC cell line was passaged for 10 successive generations. Following FMDV infection, the 146S antigen content was assessed every three generations compared with the control BHK-21 cells. Additionally, the replication stability of the mutant BHK-7 cells after suspension culture adaptation was evaluated. (**b**) BHK-7-FC cells were passaged for 10 consecutive generations, and their TCID_50_ was measured every three generations after FMDV infection, in comparison with control BHK-21 cells. The data are presented as the means ± SDs from three independent experiments and were analyzed via Student’s two-tailed unpaired *t*-tests. ***, *p* < 0.001; ****, *p* < 0.0001. T3, represents the third passage of cells, T6, represents the sixth passage of cells, and T9 represents the ninth passage of cells.

**Table 1 vetsci-12-00167-t001:** Heavy ion dose induced by ^12^C^6^ Ions (*n* = 6).

Sample	Optimal Dosage (Gy)	Optimal Dosage Rate (Gy/min)	Minimum Dose Rate Threshold (Gy/min)	Maximum Dosage Rate (Gy/min)
5 Gy	5.0	1.0	0.5	2.5
10 Gy	10.0	2.0	1.0	5.0
15 Gy	15.0	5.0	2.5	7.5
20 Gy	20.0	8.0	4.0	10.0
25 Gy	25.0	12.0	6.0	12.5
Control	0	0	0	0

## Data Availability

The raw data supporting the conclusions of this article will be made available by the authors on request.

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
