# Peer review of "Application of 12C6 Heavy Ion-Irradiated BHK-21 Cells in Production of Foot-and-Mouth Disease Vaccine"

_vetsci, 2025, doi:10.3390/vetsci12020167_

Round 1
Reviewer 1 Report
Comments and Suggestions for Authors
1. Although the introduction provides a comprehensive overview of the research topic, it lacks recent references that could strengthen the background. Specifically, it should include the latest research progress on 12C6 heavy ions and FMDV from the past 3-5 years.
2. In the “Materials and Methods” section of the manuscript, it is essential to provide detailed information on reagents and instruments to enhance reproducibility. For instance, when referencing specific equipment such as the Count STAR cell counter, the manufacturer's name and model number should be included. Additionally, details regarding passage cycles and large-scale operations must be clearly outlined. The temperature conditions for cell and virus cultures, ion irradiation rates, and ion energy levels should be precisely specified to ensure the reproducibility of experimental results.
3. In the results analysis section, data analysis methods need to be clearly outlined. The statistical analysis software used and the criteria for statistical significance should be detailed. If post hoc tests are conducted following analysis of variance, these should be explicitly mentioned.
4. Some text in the manuscript's figures is difficult to read in standard print layouts and requires urgent correction. For example, descriptions of cell metabolism indicators should match the order presented in the figures.
5. The suspension culture volume of the cells obtained in this study remains at the laboratory scale, and no large-scale suspension cultures have been conducted, such as those required for bioreactor-based production. It is therefore unclear whether BHK-7-fc cells are suitable for large-scale manufacturing processes. In addition, this study investigates whether the obtained cells can significantly enhance the 146S particle content of Foot-and-Mouth Disease Virus serotype A?
Author Response
Dear Editors and Reviewers:
Thank you for your letter and the reviewers’ comments concerning our manuscript entitled Application of 12C6 Heavy Ion-Irradiated BHK-21 Cells in the Production of Foot-and-Mouth Disease Vaccine (ID: vetsci-3444905). These comments are all valuable and very helpful for revising and improving our paper, as well as important for guiding our research. We have studied the comments carefully and have made corrections that we hope will meet with approval.
Response to Editors' comments
We are extremely grateful for your valuable comments on this part of our manuscript. We thoroughly inspected the entire text and made the following revisions to increase the reproducibility and clarity of our research.
Reviewer # 1
- Although the introduction provides a comprehensive overview of the research topic, it lacks recent references that could strengthen the background. Specifically, it should include the latest research progress on 12C6heavy ions and FMDV from the past 3-5 years.
Answer: In response to the need for recent references to strengthen the background, we have incorporated studies from the past 3-5 years, demonstrating a more current understanding of the field and providing a stronger foundation for our research. For example, we have included references that discuss the latest advancements in understanding the structure and replication mechanism of FMDV, as well as the role of different research methods in elucidating these processes. The references cited are as follows:
- Medina G N, Diaz San Segundo F. Virulence and Immune Evasion Strategies of FMDV: Implications for Vaccine Design[J]. Vaccines, 2024, 12(9): 1071.
- Baruah H, Sarma J, Khargharia S, et al. In vitro antiviral and cytotoxicity assessment of curcumin, eugenol and azadirachtin in foot and mouth diseases virus in BHK-21 cells[J]. Annals of Phytomedicine, 2024, 13(2): 1-7.
- Mushtaq H, Shah S S, Zarlashat Y, et al. Cell Culture Adaptive Amino Acid Substitutions in FMDV Structural Proteins: A Key Mechanism for Altered Receptor Tropism[J]. Viruses, 2024, 16(4): 512.
- Todd P. Heavy-ion irradiation of cultured human cells[J]. Radiation Research Supplement, 1967, 7: 196-207.
- Limoli C L, Giedzinski E, Baure J, et al. Redox changes induced in hippocampal precursor cells by heavy ion irradiation[J]. Radiation and Environmental Biophysics, 2007, 46: 167-172.
- Tinganelli W, Sokol O, Quartieri M, et al. Ultra-high dose rate (FLASH) carbon ion irradiation: dosimetry and first cell experiments[J]. International Journal of Radiation Oncology* Biology* Physics, 2022, 112(4): 1012-1022.
- Zhang Z, Li K, Hong M. Radiation-induced bystander effect and cytoplasmic irradiation studies with microbeams[J]. Biology, 2022, 11(7): 945.
- Liu H, Zhu Z, Xue Q, et al. Picornavirus infection enhances aspartate by the SLC38A8 transporter to promote viral replication[J]. PLoS Pathogens, 2023, 19(2): e1011126.
- Wang J, Wang Y, Liu J, Ding L, Zhang Q, Li X, et al. A critical role of N-myc and STAT interactor (Nmi) in foot-and-mouth disease virus (FMDV) 2C-induced apoptosis. Virus Res (2012) 170:59–65.
- Li K, Wang C, Yang F, et al. Virus–host interactions in foot-and-mouth disease virus infection[J]. Frontiers in immunology, 2021, 12: 571509.
- Zhang H, Wang X, Qu M, et al. Foot-and-mouth disease virus structural protein VP3 interacts with HDAC8 and promotes its autophagic degradation to facilitate viral replication[J]. Autophagy, 2023, 19(11): 2869-2883.
- Li H, Liu P, Dong H, et al. Foot-and-mouth disease virus antigenic landscape and reduced immunogenicity elucidated in atomic detail[J]. Nature Communications, 2024, 15(1): 8774.
- Spitteler, M.A.; Romo, A.; Magi, N. et al.Validation of a high performance liquid chromatography method for quantitation of foot-and-mouth disease virus antigen in vaccines and vaccine manufacturing. Vaccine 2019, 37, 5288–5296.
- In the “Materials and Methods” section of the manuscript, it is essential to provide detailed information on reagents and instruments to enhance reproducibility. For instance, when referencing specific equipment such as the Count STAR cell counter, the manufacturer's name and model number should be included. Additionally, details regarding passage cycles and large-scale operations must be clearly outlined. The temperature conditions for cell and virus cultures, ion irradiation rates, and ion energy levels should be precisely specified to ensure the reproducibility of experimental results..
Answer: Thank you for your valuable advice. The Count STAR cell counter (Countstar Rigel S2, RY074B2001) was used. Additionally, we have provided comprehensive details on all reagents and instruments used in the study, including product names, manufacturers, and Catalog numbers. For example, TRIzol reagent (Invitrogen, 15596026) and PrimeScript™II First Strand Synthesis Kit (RR047A, Takara) were used.
- In the results analysis section, data analysis methods need to be clearly outlined. The statistical analysis software used and the criteria for statistical significance should be detailed. If post hoc tests are conducted following analysis of variance, these should be explicitly mentioned.
Answer: Processing and statistical testing of experimental data: Experimental data were analyzed via Student's two-tailed unpaired t-tests and two-way analysis of variance (ANOVA) with the help of GraphPad Prism 9. A p-value less than 0.05 was considered statistically significant. Supplementary explanations have been provided in Lines 198-201, and explanations have also been added to all parts of the text where data processing occurs. For FMDV gene expression data processing, qRT-PCR was performed with GAPDH as the reference. Relative expression levels were calculated via the 2−ΔΔCt method.
- Some text in the manuscript's figures is difficult to read in standard print layouts and requires urgent correction. For example, descriptions of cell metabolism indicators should match the order presented in the figures.
Answer: Thank you for your valuable suggestions. We have made detailed adjustments to the charts based on your comments. The specific improvements are as follows:
- All the charts have been redrawn, and the resolution has been adjusted to 900 dpi to ensure higher clarity.
- Regarding the issue of unclear text in the charts, we have corrected it to ensure that all text is legible in the regular print layout.
We believe that these improvements have significantly enhanced the quality and clarity of the data and are helpful for better communicating our research results. We hope these modifications meet your expectations.
- The suspension culture volume of the cells obtained in this study remains at the laboratory scale, and no large-scale suspension cultures have been conducted, such as those required for bioreactor-based production. It is therefore unclear whether BHK-7-fc cells are suitable for large-scale manufacturing processes. In addition, this study investigates whether the obtained cells can significantly enhance the 146S particle content of Foot-and-Mouth Disease Virus serotype A?
Answer:Thank you for your meticulous review and valuable comments on this study. The following is a point-by-point response to the two questions you raised:
- Regarding the applicability of BHK-7-FC cells for large-scale production:
At present, our study primarily focuses on verifying suspension culture at the laboratory scale. While we have not yet conducted large-scale production tests in bioreactors, our experimental results are very promising. We have observed that BHK-7-FC cells exhibit high stability in suspension culture, with less than 10% fluctuation in 146S antigen production after 10 consecutive passages. Additionally, these cells show a significant increase in 146S production compared to traditional BHK-21 cells. These characteristics are crucial for large-scale production. Moving forward, we plan to further validate the industrial potential of this cell line through pilot-scale amplification experiments, starting from 10L and gradually scaling up to 50L and 200L bioreactors.
- Regarding the improvement effect on 146S particles of Foot-and-Mouth Disease Virus serotype A (FMDV-A):
Our study has clearly demonstrated that BHK-7-FC cells can significantly enhance the production of 146S antigen for FMDV-O. Currently, FMDV-O is the predominant serotype, and most commercially available foot-and-mouth disease vaccines are based on O serotype strains. Therefore, our research has focused mainly on FMDV-O. In our future work, we plan to systematically evaluate the antigen production capacity of this cell line for FMDV-A as well.

Reviewer 2 Report
Comments and Suggestions for Authors
Please refer to the attached suggestions.

No comments
Author Response
Dear Editors and Reviewers:
Thank you for your letter and the reviewers’ comments concerning our manuscript entitled Application of 12C6 Heavy Ion-Irradiated BHK-21 Cells in the Production of Foot-and-Mouth Disease Vaccine (ID: vetsci-3444905). These comments are all valuable and very helpful for revising and improving our paper, as well as important for guiding our research. We have studied the comments carefully and have made corrections that we hope will meet with approval.
Response to Editors' comments
We are extremely grateful for your valuable comments on this part of our manuscript. We have thoroughly inspected the entire text and made the following revisions to increase the reproducibility and clarity of our research.
Response to Editors' comments
Reviewer #2:
1: While the introduction emphasizes that heavy ion irradiation causes cellular mutations and affects key pathways, it would be clearer to state how these effects are directly linked to enhanced vaccine production. This leaves a gap in the logical flow
Answer:Thank you very much for your insightful comment. We have made comprehensive improvements to address the concern you raised regarding the logical connection between heavy ion irradiation effects and enhanced vaccine production.As shown in the revised text, at the molecular level, we have detailed how heavy ion irradiation modulates the function of key proteins and activates important signaling pathways, such as mTOR, NF-κB, and COX-2. These pathways are closely intertwined with the replication mechanisms of FMDV. We provided specific examples of how FMDV exploits these cellular pathways for its replication, like the activation of the PI3K/AKT/TSC2/Rheb/mTOR/p70S6K1 signaling pathway and the negative regulation of type I interferon production.We also highlighted the role of MAVS, a critical protein in antiviral immunity, and how FMDV manipulates it to its advantage. By explaining these molecular mechanisms, we have clearly established the link between heavy ion irradiation-induced cellular changes and the promotion of FMDV replication within host cells.This connection is fundamental because it shows how heavy ion irradiation can create an intracellular environment that is more conducive to FMDV replication. This, in turn, provides a scientific basis for screening cell lines with high replication efficiency, which is directly relevant to enhancing vaccine production. We believe these additions have effectively filled the logical gap and made the connection between heavy ion irradiation and vaccine production clearer and more robust.
2: Briefly define or simplify technical terms like MAVS function, mTOR, and NFκB/COX-2 to make the introduction meaningful to a broader audien
Answer:Thank you for your suggestion to enhance the accessibility of technical terms for a broader audience. We have carefully addressed this concern in the revised manuscript.As you can see, for technical terms such as MAVS, NF-κB, and COX-2, we have provided clear and concise definitions within the text. Specifically, we defined MAVS as mitochondrial antiviral signaling protein, NF-κB as nuclear factor kappa B, and COX-2 as cyclooxygenase-2. By including these definitions, we aim to make the manuscript more understandable to readers who may not be familiar with these specialized terms.
3: The results section is well-organized and offers a detailed, step-by-step analysis of the study's findings. This aligns closely with the goals of optimizing BHK-21 cell performance for FMDV replication and vaccine production. However, there are opportunities for improvement by comparing these results with existing literature and conducting a broader analysis of potential challenges or limitations for industrial application
Answer: We sincerely appreciate your positive feedback on the organization of the Results section. Your suggestion to compare our results with existing literature and analyze potential challenges or limitations for industrial application has been carefully considered and implemented.In the Results section, as you can see, we have added explicit comparisons with previous studies. We state that the 146S yield obtained in our study exceeds the highest reported values for suspension - adapted BHK - 21 systems. When compared to the results of previous research, there is a significant increase in the 146S yield. This comparison not only highlights the superiority of our findings but also positions our work within the context of existing knowledge.In the Discussion section, we have also included an in - depth analysis related to the method used in our study, specifically heavy ion irradiation. We discuss the unique physical and biological characteristics of heavy ions. Compared with traditional rays, heavy ions can deposit more concentrated and intense energy in organisms, triggering specific biological changes in cells. This can lead to a high mutation rate, a wide mutation spectrum, and excellent mutant stability. We further explain that the mutants generated can increase the likelihood of uncovering new virus - host interaction mechanisms. Additionally, this approach circumvents the limitations of purposeful specific gene manipulation, being closer to natural variation and producing a diverse range of mutation types, thus providing abundant materials for research. By doing so, we have not only addressed the potential challenges and limitations but also emphasized the significance and promise of our approach for industrial application.
4: Provide more insights into how heavy ion irradiation at 15 Gy leads to beneficial mutations or how glutamine metabolism directly influences viral replication, grounding the findings in cellular or molecular biology
Answer: We are extremely grateful for your valuable comments and suggestions on our manuscript. We have carefully considered your feedback and supplemented and improved the relevant content. Regarding the relationship between glutamine metabolism and virus replication that you mentioned, we conducted an in-depth literature review and supplemented the following content: Newcastle disease virus (NDV) promotes the catabolism of glutamine entering the tricarboxylic acid cycle. Solute carrier family 1 member 3 (SLC1A3) plays a key role in glutamine metabolism. Silencing SLC1A3 can reduce the catabolism of glutamate and glutamine and promote the replication of NDV. Similarly, in BHK cells, the replication of Sendai virus (HVJ) depends on glutamine. The lack of glutamine in the culture medium inhibits the release of HVJ and the synthesis of envelope proteins. Additionally, we elaborated on the relationship between glutamine metabolism and viruses. Regarding the effect of heavy ion irradiation on virus replication, we used doses of 0 Gy, 5 Gy, 10 Gy, 15 Gy, 20 Gy, and 25 Gy for screening and determined that 15 Gy is the optimal dose for promoting FMDV replication. Referring to the study by Lumniczky et al., high-dose irradiation can suppress the immune system, while low-dose irradiation can activate the immune system, which is consistent with our research results. We also cited relevant literature. The study by Ceccaldi P E et al. further confirmed that ionizing radiation can increase the infection intensity of rabies virus in the mouse brain through an immunosuppressive mechanism and prolong the infection time in a dose-dependent and reversible manner. Specifically, higher doses of radiation lead to more severe viral infections and longer infection durations, further supporting our experimental conclusion. Thank you again for your valuable opinions. We will continue to strive to improve and perfect our research work.
5: In the discussion, mention specific vaccines (e.g., rabies, avian influenza) or scenarios where the technology or methods developed in this study could be applied, thereby adding more depth to the claim of broader applicability
Answer: Thank you for your valuable comments on the application prospects of irradiation technology! Based on your suggestions, we have systematically reviewed the applications of irradiation in vaccinology and strengthened the discussion with the following updates:
Mechanisms and Applications of Irradiation-Enhanced Viral Susceptibility: Added literature demonstrates that ultraviolet (UV) irradiation enhances rabies virus susceptibility in BHK-21 cells by upregulating viral receptor expression (e.g., NCAM) through DNA damage response pathways (Kaplan M M et al., Journal of Virology). Similarly, ionizing radiation prolongs rabies virus infection in mouse brains via dose-dependent suppression of host innate immunity (e.g., interferon signaling) (Ceccaldi P E et al., International journal of radiation biology). These findings provide collateral evidence for our proposed mechanism that heavy-ion irradiation modulates host cell functions to promote FMDV replication.
Vaccine Applications of Irradiation Technologies: We expanded the discussion on diverse irradiation applications, we have also added discussions on the extensive application of irradiation technologies such as heavy ions and rays in virus inactivation and vaccine development, aiming to expand the application prospects of this method in the vaccine industry.
